# Computational Analysis of MDR1 Variants Predicts Effect on Cancer Cells via their Effect on mRNA Folding

Tal Gutman[1], Tamir Tuller [1,2]*

1 Department of Biomedical Engineering, the Engineering Faculty, Tel Aviv University, Tel-Aviv, Israel, 2 The Sagol School of Neuroscience, Tel-Aviv University, Tel Aviv, Israel

* tamirtul@tauex.tau.ac.il

## Abstract

The P-glycoprotein efflux pump, encoded by the *MDR1* gene, is an ATP-driven transporter capable of expelling a diverse array of compounds from cells. Overexpression of this protein is implicated in the multi-drug resistant phenotype observed in various cancers. Numerous studies have attempted to decipher the impact of genetic variants within *MDR1* on P-glycoprotein expression, functional activity, and clinical outcomes in cancer patients. Among these, three specific single nucleotide polymorphisms—*T1236C*, *T2677G*, and *T3435C* - have been the focus of extensive research efforts, primarily through in vitro cell line models and clinical cohort analyses. However, the findings from these studies have been remarkably contradictory. In this study, we employ a computational, data-driven approach to systematically evaluate the effects of these three variants on principal stages of the gene expression process. Leveraging current knowledge of gene regulatory mechanisms, we elucidate potential mechanisms by which these variants could modulate P-glycoprotein levels and function. Our findings suggest that all three variants significantly change the mRNA folding in their vicinity. This change in mRNA structure is predicted to increase local translation elongation rates, but not to change the protein expression. Nonetheless, the increased translation rate near *T3435C* is predicted to affect the protein's co-translational folding trajectory in the region of the second ATP binding domain. This potentially impacts P-glycoprotein conformation and function. Our study demonstrates the value of computational approaches in elucidating the functional consequences of genetic variants. This framework provides new insights into the molecular mechanisms of *MDR1* variants and their potential impact on cancer prognosis and treatment resistance. Furthermore, we introduce an approach which can be systematically applied to identify mutations potentially affecting mRNA folding in pathology. We demonstrate the utility of this approach on both ClinVar and TCGA and identify hundreds of disease related variants that modify mRNA folding at essential positions.

**Data Availability Statement:** Code can be downloaded from https://www.cs.tau.ac.il/~tamirtul/Co-Trans-FE. Non-confidential data is available on Zenodo (10.5281/zenodo.14050188) and protected data is available through the

Genomic Data Commons (GDC) with appropriate access permissions. Instructions for requesting access to GDC protected data can be found at https://gdc.cancer.gov/access-data/obtaining-access-controlled-data.

**Funding:** This study was partly funded by the Edmond J. Safra Center for bioinformatics at Tel Aviv University (a fellowship to TG) and by the Koret-Berkeley-TAU initiative in Bioinformatics and Computational Biology (to TT). The funders had no role in study design, data collection and analysis, decision to publish, or preparation of the manuscript.

**Competing interests:** The authors have declared that no competing interests exist.

## Author summary

The p-gp protein is the product of the *MDR1* gene. It functions as a cellular drug exporter and is known to be related to patients' response to chemotherapy. Our study focuses on understanding how genetic variants in the *MDR1* gene might affect the production and function of this protein. While many non-computational studies have investigated the effects of these variants, the results have been inconsistent. Using computational tools, we aim to assess how these variants influence key steps of gene expression to understand their potential impact on patients' resistance to treatment and survival. Our findings suggest that these variants, particularly *T3435C*, could alter the folding and therefore function of the p-gp protein via the effect on the folding of the mRNA, potentially affecting its role in drug resistance. Additionally, we identify variants that are predicted to potentially affect mRNA and protein folding and thus may play important roles in disease. Ultimately, we propose a framework for studying the effects of genetic variants on gene expression, which could help deepen our understanding of disease mechanisms.

## 1 Introduction

P-glycoprotein (p-gp), an ATP-driven efflux pump, was initially discovered in 1976 through its ability to modulate drug permeability in Chinese hamster ovary cells [1]. This transmembrane protein belongs to the ATP-binding cassette superfamily and is comprised of two homologous halves, each containing six transmembrane domains and an ATP binding domain [2,3]. P-gp exhibits a remarkably wide substrate specificity, capable of binding and transporting molecules that span a vast range of sizes and diverse chemical structures [4]. Physiologically, it is highly expressed at the blood-brain barrier as well as in excretory organs such as the liver, kidneys and intestine, where it functions to protect sensitive tissues and eliminate toxic compounds [5,6]. However, overexpression of p-gp in tumors confers multi-drug resistance (MDR) by enabling cancer cells to efflux chemotherapeutic agents [7].

The gene encoding p-gp is *MDR1*, located on chromosome 7. This gene exhibits considerable genetic variation, with over 50 single nucleotide polymorphisms (SNPs) identified within its coding region [8]. Among these, three specific SNPs—*T1236C (rs1128503)*, *T2677G (rs2032582)* and *T3435C (rs1045642)*—have been the subject of intense research focus over the past few decades due to their prevalence in the population. While *T1236C* and *T3435C* represent synonymous codon changes in Glycine and Isoleucine, *T2677G* results in a non-synonymous Serine to Alanine substitution. The variant alleles of these three SNPs occur in 49–57% of the population. Moreover, the *T1236C-T2677G-T3435C* haplotype is estimated to be present in approximately one-third of individuals [9].

Over the years, numerous studies have investigated the impact of these three *MDR1* SNPs on p-gp expression, function and chemotherapy response, but have reported highly contradictory findings [10]. The literature presents a complex picture, with some studies suggesting these variants lead to altered mRNA and/or protein levels, while others find no such effects [11–21]. For instance, Kimchi-Sarfaty et al. [21] and Fung et al. [20] reported no significant changes in both mRNA expression and protein abundance in their respective cell models. Similarly, Gow et al. [17] and Salama et al. [19] found no effects on either mRNA expression or protein abundance, respectively. However, other investigations [11–15] have suggested that at least some of these variants do impact mRNA expression or protein abundance under various conditions and in different tissues or cell lines. Moreover, certain studies link the SNPs to differential chemotherapy outcomes and patient survival, while others fail to find any association

[22–37]. Remarkably, even among the studies reporting survival effects, there is disagreement regarding whether the variants are beneficial or detrimental [38].

The reasons underlying these discrepancies likely stem from multiple factors. Clinical trials are inherently limited by cohort demographics like size, ethnicity, cancer type and treatment regimen. In vitro cell line models, while more controlled, cannot fully recapitulate the complexity of an in vivo cellular environment [39]. Moreover, current experimental techniques face significant challenges in accurately measuring some phases of gene expression like translation and co-translational folding. In this study, we employ diverse computational tools to model the effects of these three MDR1 variants on all major steps of gene expression. Additionally, we present a novel approach for evaluating the influence of variants on mRNA folding, demonstrating its significance in the context of *MDR1*. Furthermore, we show that this approach can be used to detect other potentially pathogenic variants that modify mRNA folding.

## 2 Methods

### 2.1. Data sources

For performing the various analyses, we utilized data of several known databases. Single Nucleotide Variants (SNV) data, mRNA expression data and clinical data of TCGA [40,41] projects was downloaded from the GDC [42] (https://gdc.cancer.gov/) on November 2021. Additional clinical data of the same patients was downloaded from cBioPortal [43] (https://www.cbioportal.org/) on October 2024. The complete human coding sequence (CDS) was downloaded from Ensembl [44,45] (https://ftp.ensembl.org/pub/release-109/fasta/homo_sapiens/cds/) on 2020. Human protein expression measurements were downloaded from PaxDb [46] on 2020 (https://pax-db.org/dataset/9606/1502934799/). The positions of the ATP binding domains of MDR1 were taken from Uniprot [47]. SNV and indel data from the 1000 genomes project [48] was downloaded on October 2023 using the ftp server (http://ftp.1000genomes.ebi.ac.uk/vol1/ftp/data_collections/1000_genomes_project/release/20190312_biallelic_SNV_and_INDEL/). All ClinVar [49] variants were downloaded from the website on April 2023 (https://ftp.ncbi.nlm.nih.gov/pub/clinvar/). CATH3D protein domain annotations were downloaded from InterPro [50] on September 2023 using their API.

### 2.2. *MDR1* expression

**MDR1 *expression change of TCGA patients*.**   SNVs and expression data of TCGA patients were used to examine the effects of *T1236C*, *T2677G*, *T3435C* and the haplotypes on *MDR1* expression. For each mutation or haplotype, the cohort was split to carriers and non-carriers groups. The non-carriers group was randomly sampled 100,000 times such that each sampled group contained the same amount of patients as the carriers group. The average *MDR1* expression was calculated for each sampled group and used to create a distribution of the *MDR1* expression for non-carriers. The average *MDR1* expression of the carriers group was compared to the distribution and an empirical p-value was calculated.

**MDR1 *expression change according to Enformer*.**   Enformer [51] is a state-of-the-art model that predicts gene expression and transcription regulation based on exceptionally long input sequence (hundreds of thousands of nucleotides). It generates predicted measurements for 5,313 tracks, varying across different experiments and tissue types. Enformer was used to predict *MDR1* expression levels both for the wild type and the mutated sequence, for all three variants. When subtracting the output matrices of the wild type and mutant sequences we capture the difference in transcription regulation caused by the variant. To map this 2-dimensional difference matrix to a single score we try four different approaches, each time

calculating the mean of the difference using a subset of the tracks. We use: [1] all tracks. [2] only cap analysis of gene expression (CAGE) tracks, as they predict gene expression more directly than other experiments predicted in the Enformer output. [3] tracks associated with tissues where MDR1 is highly expressed. [4] CAGE tracks where MDR1 is highly expressed.

The same process was performed for random variants with similar characteristics (explained in the "Empirical p-values" section) and the scores of the original and random variants were compared. If the score of the variant was larger than the scores of 95% of the random variants it was deemed to significantly change *MDR1* expression.

## 2.3. Post-transcriptional regulation

**Splicing.**   To examine the effect of *T1236C*, *T2677G*, *T3435C* on the occurrence of splicing events we used SpliceAI [52], a model that, given a genomic sequence of length 11,001, predicts the probabilities of each site to be a donor or acceptor site. We perform the prediction for both the reference and mutated sequence, centered around each of the three variants. For each position in the prediction range (the 1001 central nucleotides), we calculated the difference in the probability of it being a donor/acceptor site that was caused by the variant. We searched for positions for which the probability changed by more than 50%. Positions exhibiting an increment of more than 50% were considered new prospective donor/acceptor sites, whereas positions for which the probability decreases by more than 50% are regarded as potentially abolished donor/acceptor sites.

**A-to-I editing.**   Adenosine-to-inosine (A-to-I) editing occurs in double-stranded RNA structures and influences mRNA stability, splicing and translation. To assess how the variants impact this editing process, we utilized Airliner [53], a linear regression based tool that predicts the probability of A-to-I editing at specific nucleotide positions. The input consists of a 1001-nucleotide sequence centered around the variant position, and the output is a probability vector, indicating the editing probability of each adenosine within the input sequence. Adenosine positions exhibiting a change larger than 50% between the reference and mutated sequence were considered modified A-to-I sites.

**m6A.**   N6-methyladenosine (m6A) is another common post-transcriptional modification in eukaryotes, where the N6-position of adenosine is methylated. To find whether the variant modify m6A sites, we used DeepM6ASeq [54]. The input to this deep learning based tool is an RNA sequence (we used the maximum length enabled by the model, 101 nucleotides) and the output is a list of predicted m6A sites and their respective probabilities. Adenosine positions exhibiting a change larger than 50% between the reference and mutated sequences were considered modified m6A sites.

**Interaction with RNA-binding proteins (RBPs).**   To determine whether the variants alter the *MDR1* mRNA interactions with RBPs, we used catRAPID omics [55]. This model integrates data on mRNA secondary structure, hydrogen bonding, and van der Waals forces to predict the interaction probability between protein-mRNA pairs. We examined the z-scores of these interactions for both the reference and mutated sequences to identify sites where binding has significantly changed. The z-score represents the normalized interaction probability, and a site was considered significantly modified if the difference in z-scores exceeded 1, indicating that the mutation increased or decreased the normalized binding probability by more than one standard deviation compared to the reference.

## 2.4. Local translation rates

To examine the effect of *T1236C*, *T2677G* and *T3435C* on translation rates in their vicinity we utilized several measures positively correlated with it—MFE, CAI, FPTC and tAI. We

calculated these measures for both the reference and mutated MDR1 CDS sequences and examined the difference in these measures at the position of the variant.

**MFE.** A per-position MFE score was computed using ViennaRNA [56]; first, a sliding window (length = 39 nucleotides, stride = 1 nucleotide) was used to obtain a per-window MFE score. Then, the MFE score of a specific position in the CDS was set as the average of all MFE scores of the windows that the position is in. The secondary structure of the mRNA near the variants was also predicted by ViennaRNA and visualized using forna [57].

**CAI.** Human CAI weights were computed as suggested in the original paper [58]. The set of highly expressed genes (15% most highly expressed) was curated using human protein expression levels from PAXdb [46].

**FPTC.** Human FPTC weights were downloaded from the Kazusa website [59].

**tAI.** tAI tissue-specific weights were taken from Hernandez-Alias et al. [60] and the s weights were optimized as depicted in Sabi et al. [61]

## 2.5. Global translation rates

To understand whether the variants' effect on local translation rates have an impact on the global translation rate and on p-gp expression, we utilize the totally asymmetric simple exclusion process (TASEP) model [62,63]. As the codon elongation rates are proportional to the exponent of the MFE [63], we estimated them according to the following formula–

$$\lambda_i = e^{0.1*MFE_i} \tag{1}$$

Where $MFE_i$ is the MFE score of codon i and $\lambda_i$ is its elongation rate

We ran the model for different initiation rates ($\lambda_{init}$ = 0.05, 0.3, 0.6, 1) and for the reference and variant sequences. We assured we ran the model long enough to reach a steady state. For each combination of parameters, we repeated the simulation one-hundred times, and saved the protein production rate and site densities. A Wilcoxon rank sum test was used to compare the protein production rates of the reference and variant sequences.

## 2.6. Co-translational folding

**Model development.** To examine the effect of *T1236C*, *T2677G* and *T3435C* on the CTF of p-gp, a computational model that assesses which positions in the CDS are important for correct protein folding was deployed. The model is yet to be published and therefore we will provide much detail about the model's methodology (full description in S1 Text). The analysis is based on the basic assumption that CTF is governed by local translation rates, and that this rate is evolutionary conserved for a position that is crucial for correct folding [64]. Thus, it searches for positions with both evolutionarily conserved low and evolutionarily conserved high MFE (a measure correlated with translation rate) across orthologous versions of a gene. All *MDR1* orthologous CDSs (n = 149) were downloaded from Ensembl (https://rest.ensembl.org/documentation/info/homology_ensemblgene) and were aligned using Clustal Omega [65]. The MFE score per nucleotide position was calculated for all sequences in the multiple sequence alignment (MSA). Then, we calculated the average MFE at each CDS position across the different organisms. To find the positions where the MFE score is conserved as significantly lower or higher than expected by chance, we created two kinds of permuted versions of the MSA. In the first method (named "vertical permutation"), we shuffle synonymous codons within the same column of the MSA. In the second method (named "horizontal" permutation), we horizontally swap between synonymous codons of pairs of columns. Both methods affect the MFE scores while keeping fundamental characteristics of the MSA such as the evolutionary distances between organisms and the MSA score. One hundred permutations of the

MSA are created for each kind. We calculated the per-position MFE score averaged across orthologs for each permuted MSA in the same manner as was done for the original one. At this point, for each CDS position we have a single true MFE score and one hundred scores from each of the permuted versions. We utilized the permutations to calculate a z-score for each position (see Eq 2). Finally, we intersect the results to get positions that had significantly low or high MFE when compared to both kinds of permuted versions of the MSA.

$$z_i = \frac{MFE_i - \mu_i}{\sigma_i} \qquad (2)$$

Where $MFE_i$ is the MFE score of position i in the original sequence and $\mu_i$, $\sigma_i$ are the mean and standard deviations of the MFE scores of position i in all the permuted versions of the sequence.

**Model evaluation.** To show that the z-scores outputted by the model are meaningful and can be used to find CTF altering variants we utilize variant information from both ClinVar and TCGA. For both ClinVar and TCGA, we keep only SNPs or short indels that are in the coding region. For ClinVar we also keep only variants that are labeled as benign or pathogenic, removing variants with more ambiguous definitions. Then, we obtain a z-score for each variant according to its genomic positions and group the variants according to their z-scores. For the ClinVar analysis we examine the pathogenic/benign ratio in each group and for TCGA we examine the percent of variants that are also found in the 1000 genomes project in each group.

**CTF modifying variants.** To obtain a list of TCGA/ClinVar variants with the highest potential of affecting CTF we define the following criteria:

- In positions that receive the bottom or top 5% of the MFE z-scores in the database

- Cause a change in MFE that is equal or larger than 1.5

- Less than 200 nucleotides away from a structural domain border. Information regarding domains was taken from Interpro [50].

- For TCGA variants only—are prevalent in more than 1% of patients for at least one cancer type, or have a higher frequency than 1% across TCGA, and occur in more than a single patient

**Functional enrichment analysis.** Gorilla [66] was used to compare the genes that contained CTF modifying variants to a background set of genes. For TCGA, the background set was all the human protein coding genes which have mutational information on TCGA. For ClinVar variants, the background set was all the human protein coding genes which have mutational information on ClinVar. We searched for enriched functions, processes and components and demanded a corrected p-value smaller than 0.05.

### 2.7. Survivability of TCGA patients

To examine the effect of *T1236C*, *T2677G*, *T3435C* and their haplotypes on the survival of cancer patients we used TCGA SNV and clinical data. We used the vital status of the patients and a matching time-stamp in order to create Kaplan-Meier overall survival curves [67] for the carriers and non-carriers groups. The time-stamp was derived from the maximum value of the following attributes- "Days from the initial diagnosis to current follow-up", "Days from the initial diagnosis to the current confirmation of vital status", "Days from the initial diagnosis to patient death" and the vital status from the attribute "days_to_death". For the progression free survival curves we use the attributes "Progression Free Status" and "Progress Free Survival (Months)". The logrank test [68] was used to assess whether the survival curves of the carriers

and non-carriers group significantly differ. To assert that the results are not caused by differences in tumor mutational burden (TMB), we repeated the analysis when controlling for this confounder; the balance of the TMB between the mutated and control groups was tested using a two sample KS test [69]. The analyses depicted in the Results section are conducted on the pan-cancer cohort of patients, from all TCGA cancer types. The analyses depicted in supplemental file 1 stratify patients according to groups of cancer types.

## 2.8. Stratification of TCGA patients

For several analyses TCGA patients were stratified according to their cancer types as suggested by Gao et. al [70] (see S1 Table). The patients were assigned to one of three groups–metabolic cancers, which are associated with altered metabolic pathways, proliferative cancers, which are associated with dysregulated cell proliferation, and inflammatory cancers, which are associated with immune system dysregulation. Because cancers in these categories dysregulate different pathways, it is likely that drug resistance patterns also differ between these categories. For example, metabolic cancers could dysregulate drug-metabolizing enzymes or drug efflux pumps, while proliferative cancers have rapid division rates and could lead to emergence of drug resistant clones through acquisition of new mutations.

## 2.9. Empirical p-values

To infer the significance of the changes caused by *T1236C*, *T2677G* and *T3435C* in several of the analyses, they were compared to changes caused by random variants with similar characteristics. T2677G is a non-synonymous variant and therefore its effect was compared to other, randomly sampled, T->G non-synonymous variants in the *MDR1* CDS. *T1236C* and *T3435C* are synonymous variants and therefore were both compared to randomly sampled synonymous T->C variants in the MDR1 CDS. Each original variant was compared to one-hundred randomized sequences.

## 2.10. TCGA variants correlated with T1236C

*T1236C* was found as correlated with worse overall survival. To investigate the possibility of a causal relationship, we examined mutations that are highly correlated with *T1236C* and their effect on patient survival, searching for other potential effects. Highly correlated variants were defined as variants that are detected in the genomes of more than 75% of *T1236C* positive patients.

## 3. Results

Various computational approaches were utilized to investigate the potential effects of the *T1236C*, *T2677G*, and *T3435C* variants on distinct stages of the gene expression pathway (see S1 Fig). To achieve this objective, we also obtained genetic information, clinical records, and gene expression data curated from The Cancer Genome Atlas (TCGA) repository. Importantly, one must distinguish between germline variants, which are inherited genetic variations that may be common in the population, and somatic variants, which represent genetic changes that arise within an individual's lifetime. The majority of prior investigations on MDR1 have focused primarily on germline variants. However, our study takes a broader approach, encompassing both germline and somatic variants. Specifically, when analyzing data from TCGA, we strictly assess the impact of somatic variants present in cancerous tissues. Conversely, the remaining computational analyses, which do not rely on TCGA data, are agnostic to the

variant type and can be employed to elucidate the effects stemming from the nucleotide alteration itself, irrespective of whether it originated as a germline or somatic variant.

### 3.1. *T1236C*, *T2677G*, and *T3435C* variants are not predicted to modify mRNA expression

An analysis was performed to compare the *MDR1* expression levels in TCGA patients who harbored any of the three somatic variants under investigation with those who did not (see Methods). The findings, depicted in Fig 1, suggest a borderline association between the *T1236C* variant and *MDR1* expression levels (p = 0.07), while no significant association was observed between the *T2677G* and *T3435C* variants and *MDR1* expression. Moreover, the analysis revealed no significant differences in mRNA expression levels between individuals carrying the haplotypes and those who did not (S2 Fig). Upon stratifying patients by cancer types (S1 Table), we observed no significant changes in MDR1 expression levels associated with any of the variants across different cancer categories. (S2 Table).

Moreover, we employed the computational tool Enformer [51] to evaluate the likelihood of these variants influencing *MDR1* expression levels. Enformer is a transformer-based

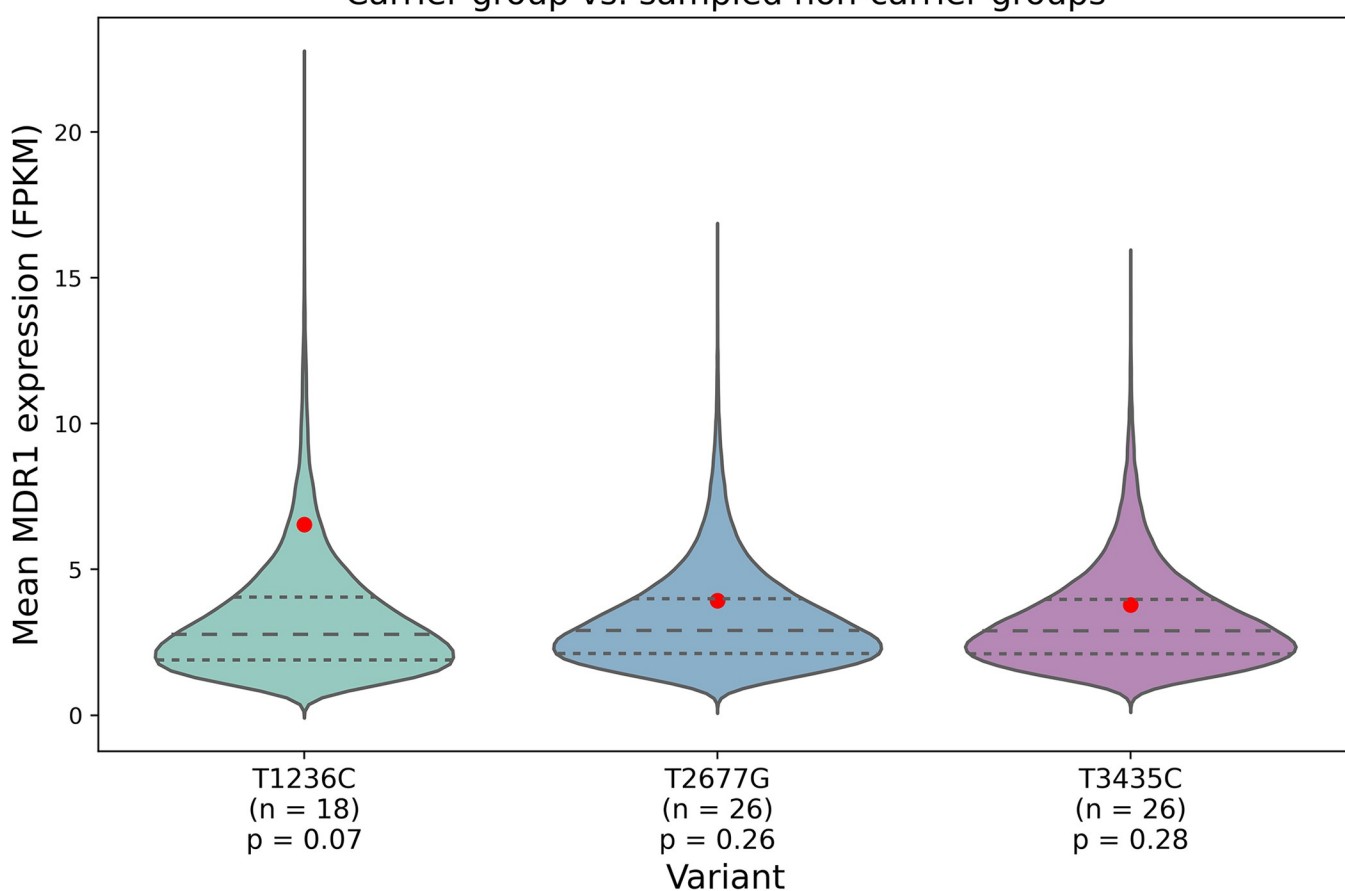

**Fig 1. Comparison of *MDR1* expression in TCGA patients with and without the studied variants.** a) *T1236C*, b) *T2677G*, c) *T3435C*. Red dots represent mean *MDR1* expression in variant carriers. Violin plots show distribution of mean expression in 100,000 randomly sampled non-carrier groups. Sample sizes (n) for carrier and non-carrier groups are indicated below each plot.

neural network model trained on extensive epigenetic and transcriptional datasets, enabling it to predict gene expression patterns. Notably, Enformer boasts the widest receptive field among gene expression predictors, analyzing input sequences spanning 393,216 nucleotides. For each variant under investigation, we ran Enformer on both the reference sequence and the mutated sequence (see Methods for details). The predicted changes in *MDR1* expression levels were not statistically significant for any of the variants, including *T1236C*.

Considering the results of both the TCGA and Enformer analyses and the notion that all three variants are far from the transcription start site (TSS) and are not in known enhancer regions, there is no strong evidence that suggests that any of the three variants affect mRNA expression.

### 3.2. *T1236C, T2677G* and *T3435C* variants are not predicted to impact post-transcriptional modifications

We examined the effect of the variants on several fundamental mechanisms of post-transcriptional regulation–splicing, A-to-I editing, m6A methylation and binding of RNA-binding proteins (RBPs) and applied a similar methodology for all cases. We used predictive models to predict positions of splice sites (SpliceAI [52]), A-to-I editing sites (AIRliner [53]), methylation sites (DeepM6ASeq [54]) or RBP binding sites (catRAPID omics [55]). We predicted both for the reference and the mutated sequence for each variant and determined whether any of the sites have been modified (see Methods for more details). The results suggest that the variants are unlikely to modify these mechanisms of post-transcriptional regulation.

### 3.3. *T1236C, T2677G*, and *T3435C* variants are predicted to increase local translation rates but not to significantly change p-gp abundance

Among other factors, protein abundance is significantly influenced by the rate of translation elongation. Even a single codon's translation rate alteration can affect protein levels if it occurs within a critical translation bottleneck. The translation rate fluctuates along the mRNA sequence and is influenced by various cellular factors, including mRNA accessibility to ribosomes, codon usage preferences, and tRNA availability.

In this section, we analyze several metrics associated with translation rate and assess whether the three variants under study are likely to modify translation efficiency, either locally or globally. Our aim is to understand how these genetic changes might impact the production of p-glycoprotein (p-gp) by potentially altering the translation dynamics of the *MDR1* gene.

**The *T1236C, T2677G*, and *T3435C* variants all reduce mRNA folding strength and alter mRNA secondary structure.** Minimum Free Energy (MFE) is a metric used to predict mRNA secondary structure. Lower MFE values typically indicate a more compact structure that is less accessible for translation, thus correlating with reduced translation rates. We calculated MFE scores for the regions surrounding each variant (see Methods) to predict changes in MFE and secondary structure.

The results (Fig 2) demonstrate that all three variants induce an increase in MFE, indicative of reduced folding of the mRNA secondary structure and potentially higher local translation rate. These observed speed elevations are statistically significant (p = 0.01, 0.02, and 0.04 for *T1236C, T2677G*, and *T3435C* respectively) when compared to MFE changes caused by random variants with similar characteristics in the *MDR1* gene (see Methods).

**T1236C, T2677G and T3435C optimize codon usage.** To assess the impact of the three variants on codon usage bias (CUB), we employed two metrics- Codon Adaptation Index

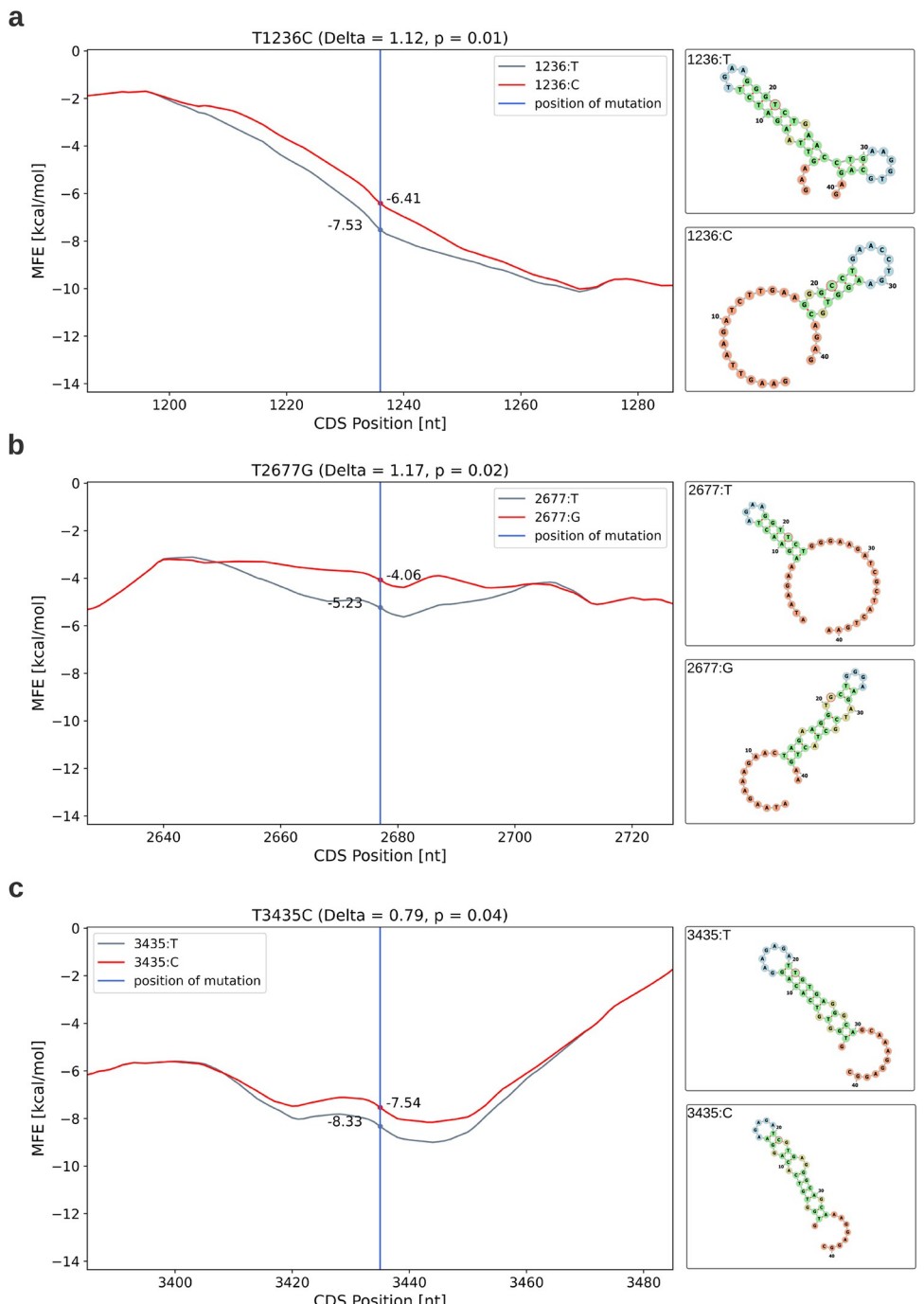

**Fig 2. Impact of variants on MFE and mRNA secondary structure.** a) *T1236C*, b) *T2677G*, c) *T3435C*. Left: MFE profiles near variant sites. X-axis shows CDS position, Y-axis shows MFE score. Blue vertical line marks variant position. Grey curve: reference sequence MFE; Red curve: mutated sequence MFE. Right: Predicted mRNA secondary structures. Top: reference sequence; Bottom: mutated sequence. Color-coded nucleotides indicate structural elements: stems (green), junctions (red), interior loops (yellow), hairpin loops (blue). Variant position is outlined in red.

(CAI) for the synonymous variants and Frequency Per 1000 Codons (FPTC) for the non-synonymous variant. Higher CUB values typically indicate better adaptation to cellular gene expression machinery and resources, correlating with increased translation rate.

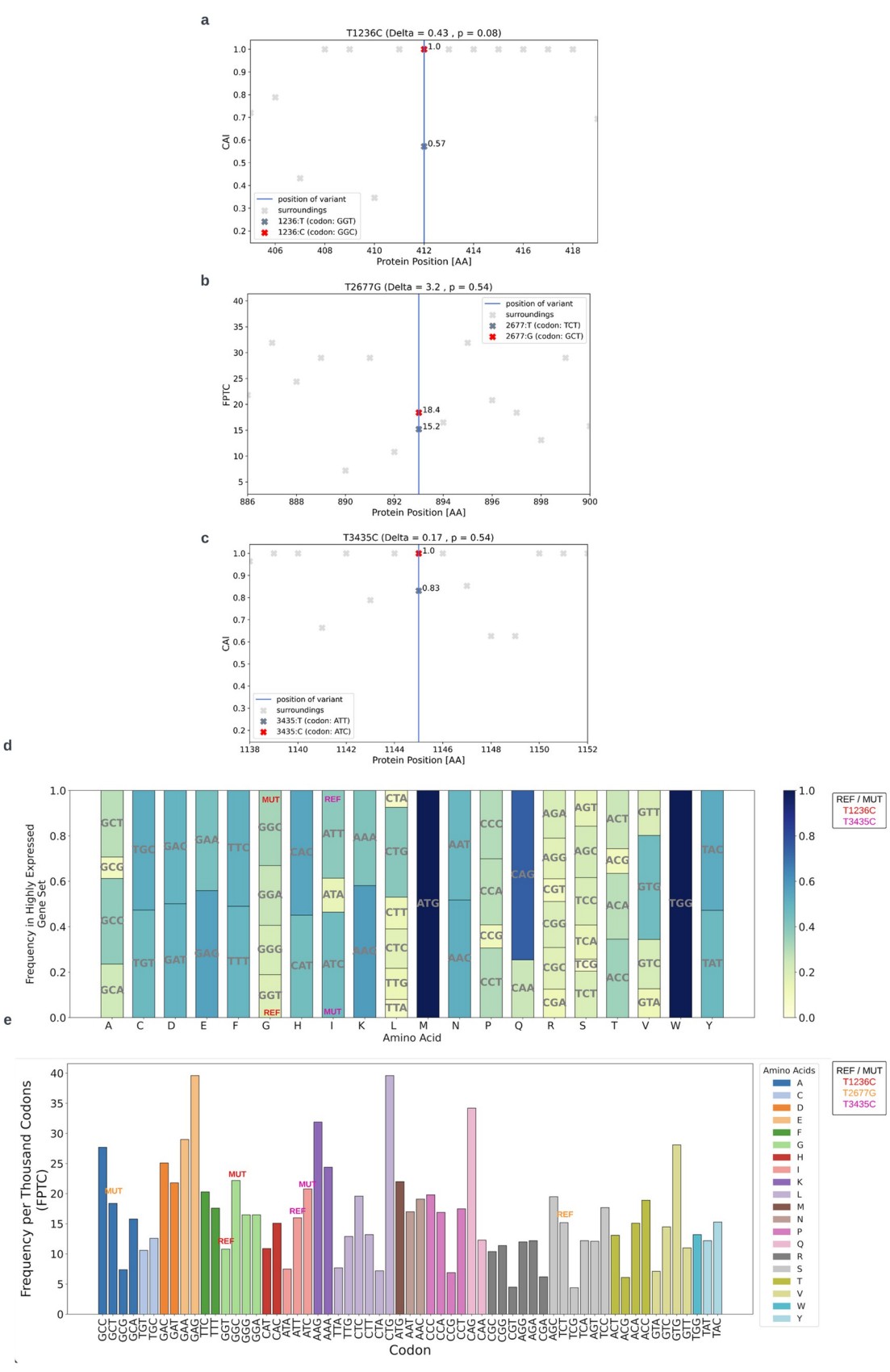

**Fig 3. Impact of the three variants on CUB.** a) *T1236C*, b) *T2677G*, c) *T3435C*. The x-axis represents the amino acid position in the protein sequence, and the y-axis represents the CUB score (CAI/FPTC). The vertical line denotes the position of the variant. Light grey x's mark the CUB scores near the variant, while the x marks on the vertical line represent the CUB score at the variant position, for the reference (dark grey) and mutated (red) codon. (d) Distribution of synonymous codon usage for each amino acid in the set of highly expressed genes. Codons before and after each variant are marked as "REF" and "MUT" respectively, colored in red (*T1236C*) or fuchsia (*T3435C*). (e) Distribution of all codons in the human genome, with bar colors indicating the amino acid encoded by the codon. Codons before and after each variant are marked as "REF" and "MUT" respectively, colored in red (*T1236C*), orange (*T2677G*), or fuchsia (*T3435C*).

Our analysis (Fig 3) reveals that all three variants result in the replacement of less common codons with more prevalent ones, suggesting a localized increase in translation rate. Notably, the *T1236C* variant substitutes the least frequent codon for Glycine with the most common one. When compared to random variants with similar characteristics in the MDR1 gene, the increase in CAI caused by *T1236C* approaches statistical significance (p = 0.08).

**T1236C and T2677G variants enhance adaptation to the tRNA pool in MDR1-expressing tissues.** The tRNA Adaptation Index (tAI) is a measure of translational efficiency that accounts for intracellular tRNA concentrations and the effectiveness of codon-anticodon pairings. Higher tAI scores indicate better tRNA availability and are associated with increased translation rates. As tRNA availability varies significantly across different tissues and organs, we analyzed the impact of the variants on tAI in tissues where *MDR1* is typically expressed: liver, kidney, colon, and brain. tAI scores for adrenal glands were not available.

As shown in Fig 4A, both *T1236C* and *T2677G* lead to increased tAI scores in all four examined tissues. The *T2677G* variant is particularly noteworthy, as it replaces a codon with extremely low tRNA availability with one that has much higher availability across all studied tissues (Fig 4C). The original codon (TCT) potentially acts as a translation bottleneck, falling within the 1st-5th percentile of tAI scores (depending on the tissue) among all codons in the MDR1 coding sequence.

Interestingly, the codons surrounding the *T2677G* variant site generally exhibit much higher tAI scores. For example, Fig 4B illustrates the tAI scores of nearby codons in epithelial colon cells, emphasizing the inefficiency of the original codon within the local translational context.

**T1236C, T2677G, and T3435C are not predicted to effect global translation rates and p-gp expression.** In the previous parts of this section, we examined several distinct measures that correlate with translation rates. All examined measures suggest that the three variants cause an increase in the local translation rates in their vicinity. To understand whether these local rate changes affect the global translation rate and p-gp expression, we utilize the TASEP (totally asymmetric simple exclusion process) model to simulate ribosome movement on the reference and variant sequences of MDR1 (See Methods). Though all three variants are predicted to cause an increase in local translation rates, we can see that they are not predicted to be the elongation bottleneck; the minimum elongation rate is detected at amino acid 1067, between *T2677G* and *T3435C* (Fig 5A). Next, we examine the effect of the variants on the ribosomal densities predicted by the model; Ribosomal density is the probability of a codon to be occupied by a ribosome at an arbitrary moment in time. Higher densities indicate the site is more often occupied by a ribosome and implies the site is more slowly translated. we see that while all three variants cause small changes in densities of various codons, overall, the density profile remains very similar to the one of the reference sequence (Fig 5B). Namely, the density is high before the translational bottleneck at position 1067 and drops sharply after it. This profile is an indication that, given that the initiation rate is high enough, the bottleneck at 1067 causes a "traffic jam" and is the dominant factor controlling the protein production rate rather than the rate at the variants' positions. Indeed, when we examine the predicted protein production rate, we see extremely small and mostly non-significant changes (Fig 5C).

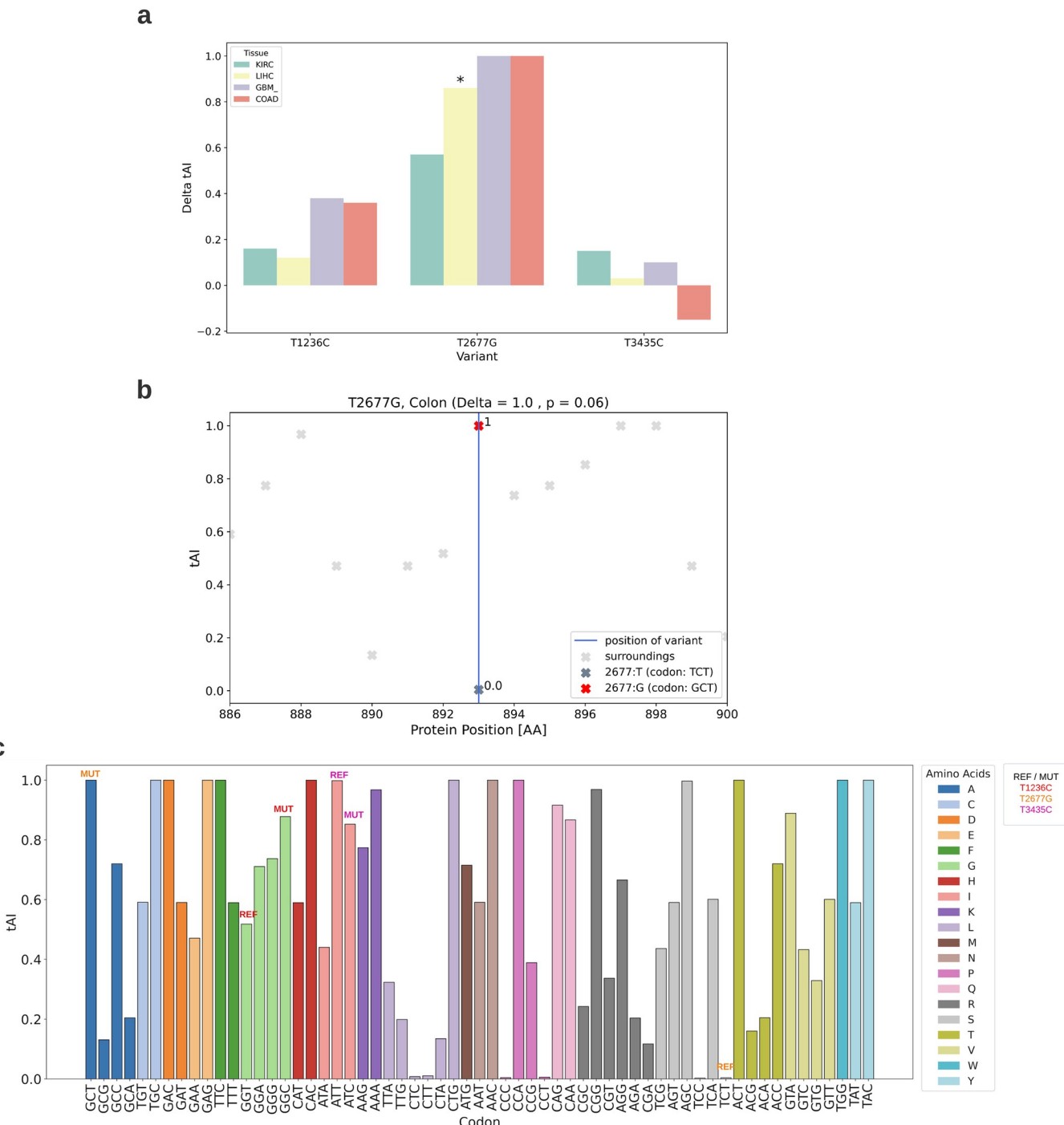

**Fig 4. Impact of the three variants on tAI.** a) Changes in tAI caused by the three variants in tissues where *MDR1* is highly expressed. Variants are shown in green for kidney renal clear cells (KIRC), yellow for liver hepatocytes (LIHC), purple for astrocytes (GBM), and red for epithelial colon cells (COAD). b) Change in tAI caused by *T2677G* in epithelial colon cells. The vertical blue line indicates the position of the variant. The tAI scores of the original and mutated codon are shown with dark grey and red x marks, respectively, while light grey x marks indicate the tAI scores of the surrounding codons. c) tAI scores of all codons in human epithelial colon cells, with bar colors representing the amino acid encoded by each codon. Codons with or without each variant are marked as "REF" and "MUT" respectively, colored in red (*T1236C*), orange (*T2677G*), or fuchsia (*T3435C*).

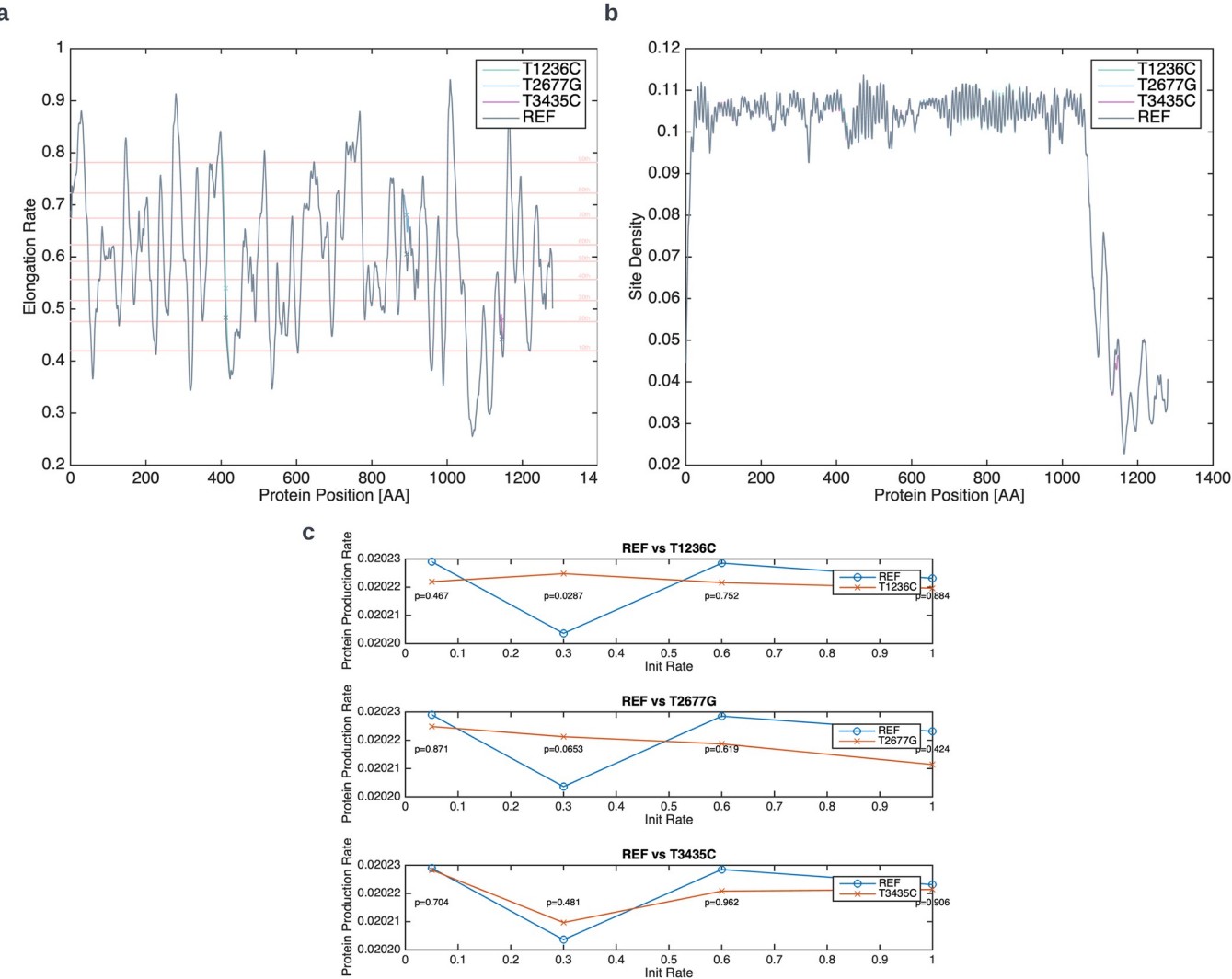

**Fig 5.** *MDR1* **translation simulation using the TASEP model.** a) Estimated elongation rates for each codon in the CDS of the reference and variant sequences (input of the model). Pink horizontal lines denote the percentiles. "x"s on the curves mark the positions of the variants. b) Density of each site (codon) at an arbitrary point in time for the reference and variant sequences. c) Protein production rates for different initialization rates. Each subplot depicts the reference sequence vs. one of the variant sequences. P-values were computed using Wilcoxon rank sum test.

### 3.4. *T1236C* variant potentially associated with decreased patient overall survivability

To evaluate the impact of the variants on patient survival, we analyzed clinical data from TCGA. We conducted overall survival (OS) and progression-free survival (PFS) analyses by constructing Kaplan-Meier curves to compare between groups of patients who developed at least one of the three variants in their tumor tissue, and those who did not (see Methods).

Our analyses (Figs 6 and S3 for haplotypes) suggest that the *T1236C* variant is associated with decreased overall survival probability (logrank statistic = 5.22, p = 0.02; Bonferroni corrected p = 0.06) but shows no significant impact on progression-free survival. Conversely, *T2677G* is linked to improved progression-free survival, while having no significant effect on overall survival. *T3435C* did not show significant associations with survival outcomes.

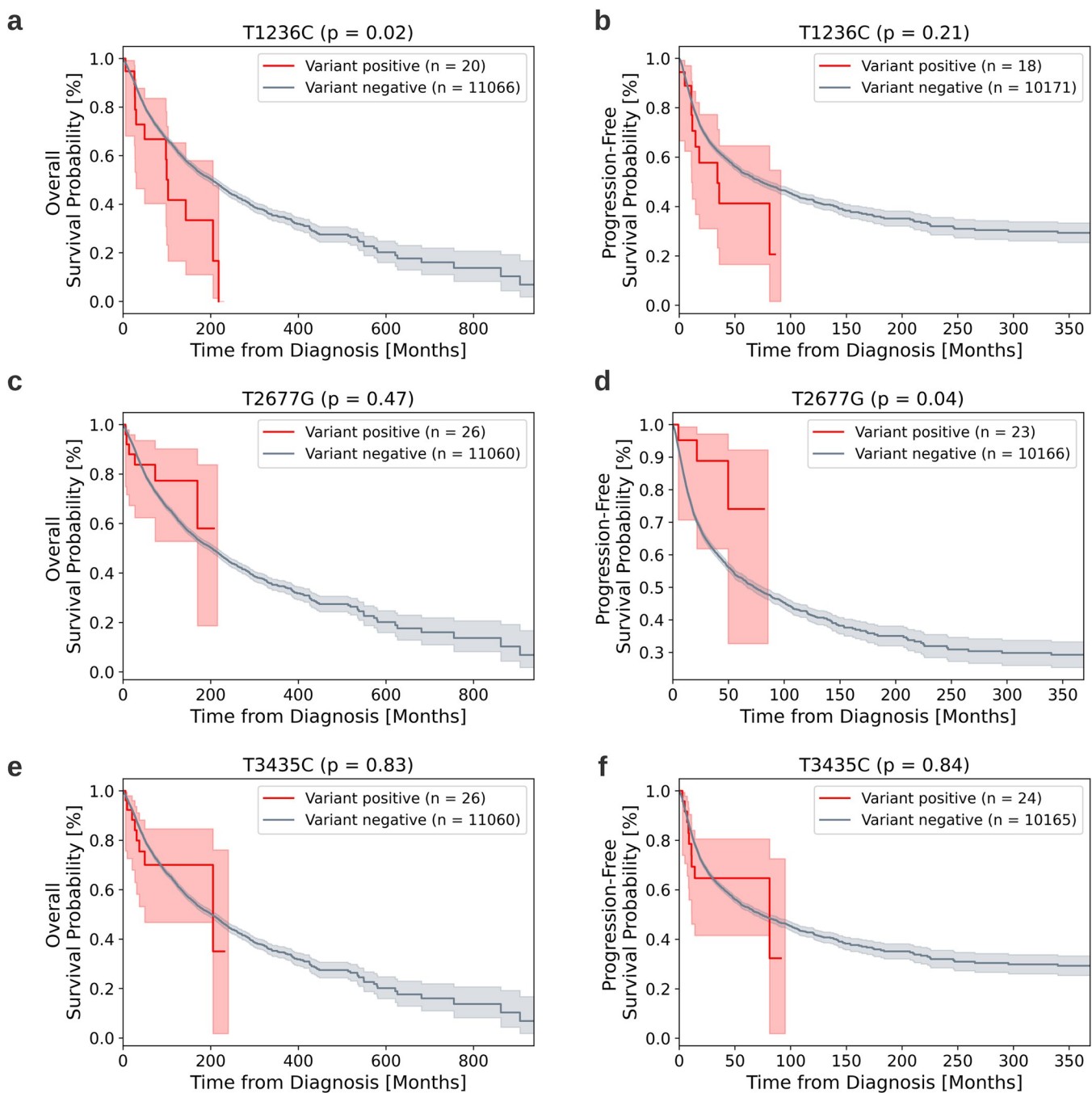

**Fig 6. Impact of the three variants on patient survival.** a-b) *T1236C*; c-d) *T2677G*; e-f) *T3435C*. Left column: overall survival. Right column: progression-free survival. Comparison of the Kaplan-Meier survival curves of the variant positive group (red) and the variant negative group (gray).

To further validate the specificity of this finding, we examined other TCGA variants highly correlated with the presence of *T1236C* (see Methods). These correlated variants did not show associations with decreased overall survivability (S4 Fig), strengthening the possibility that *T1236C* itself may be a causal variant affecting patient outcomes.

When stratifying patients by cancer types (S1 Table), we observed a significant negative association of *T1236C* with the overall survival (p = 0.002) and progression-free survival

(p = 0.01) for patients with inflammatory cancers (S5 and S6 Figs). This finding suggests that the effect of the *T1236C* variant may be particularly pronounced in certain cancer subtypes.

These results indicate that the *T1236C* variant in the MDR1 gene could potentially serve as a prognostic marker, especially in inflammatory cancers. The observed association with decreased survivability aligns with the hypothesis that this variant may enhance p-glycoprotein function, potentially leading to increased drug resistance in cancer cells. However, it's important to note that further research is needed to establish a direct causal relationship and to understand the precise mechanisms by which this variant might influence patient outcomes in different cancer contexts.

### 3.5. *T3435C* variant potentially alters co-translational protein folding by increasing local translation rate in a conserved slowly translated region within the second ATP binding domain

Co-translational folding (CTF) is the process whereby a protein begins to fold into its three-dimensional structure while still being synthesized by the ribosome. The folding segments are those that have recently emerged from the ribosome exit tunnel. The translation rate of codons being translated concurrently with the emergence of new peptide segments can significantly impact this folding process. Consequently, variants that alter local translation rates, like *T1236C*, *T2677G* and *T3435C*. may influence the protein's folding trajectory.

To investigate whether the three variants affect CTF, we created a model that identifies positions with evolutionarily conserved extreme MFE scores. This approach is based on the premise that MFE scores can serve as proxies for translation rates, and positions with evolutionarily conserved extreme translation rates (particularly slow rates) are likely maintained due to their importance in optimal CTF. Thus, variants in these positions might interfere with the CTF mechanism.

To identify positions in the *MDR1* CDS with conserved extreme MFE scores, the model analyzes orthologous *MDR1* genes from hundreds of organisms, calculates their MFE profiles, and compares them to MFE profiles of permuted versions of these genes to obtain a z-score for each position (see Methods for details).

The model's output (Fig 7) reveals that *T3435C* is located in a position with evolutionarily conserved low MFE, within a stretch of positions exhibiting conserved low MFE (6 nucleotides upstream and 17 downstream of *T3435C*). Combining these results with our previous analysis, suggesting that *T3435C* causes a local increase in translation rate, we infer that *T3435C* may increase the translation rate in a region of conserved slow translation, potentially modifying the CTF process. Notably, neither *T1236C* nor *T2677G* were found to be in positions of evolutionarily conserved extreme MFE.

This analysis suggests that the *T3435C* variant may have implications for the co-translational folding of p-glycoprotein. Such alterations in protein folding could potentially impact the function or stability of p-glycoprotein, which may contribute to the variant's association with drug resistance in cancer cells.

The significance of the *T3435C* variant's location becomes particularly apparent when considering its position within the second ATP binding domain of p-gp. This domain, typical of many ATP/GTP binding domains, adopts a P-loop NTPase fold comprising several conserved structural motifs (as illustrated in Fig 8A).

Previous research has shown that translational pauses often occur within protein domains, effectively separating structural motifs. These pauses are thought to facilitate proper folding of individual motifs. In light of this, the conserved region of low MFE where *T3435C* is situated

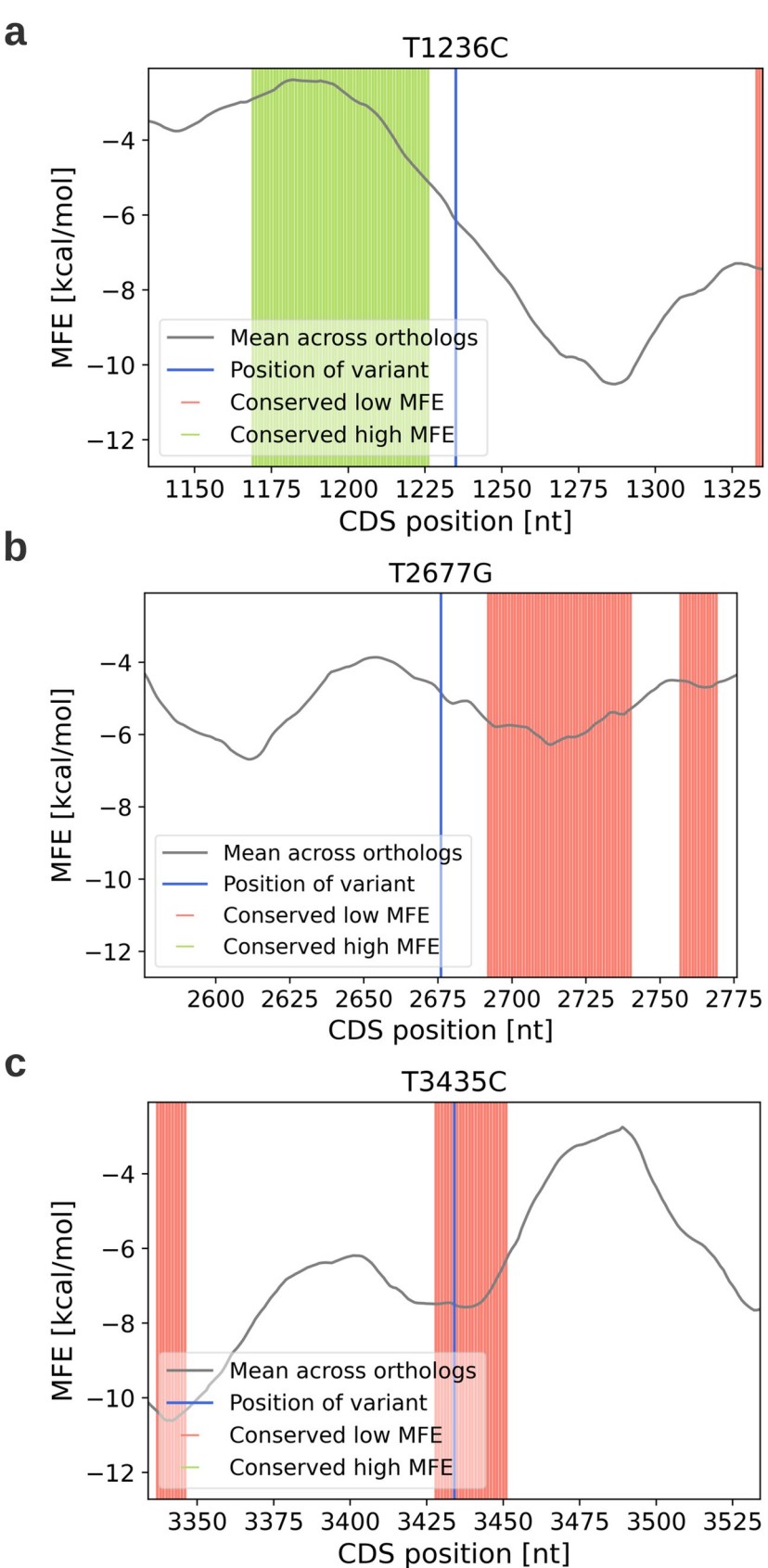

**Fig 7. Conserved regions with extreme MFE near the three variants.** a) *T1236C*, b) *T2677G*, c) *T3435C*. The x-axis denotes the nucleotide position within the CDS of the *MDR1* gene, while the y-axis shows the MFE score, which serves as a proxy for translation rate in this model. The grey curve represents the average MFE score of the *MDR1* CDS across 383 orthologs. Green horizontal lines highlight positions with conserved high MFE across different organisms, whereas red horizontal lines indicate positions with conserved low MFE across these organisms.

may play a crucial role in ensuring the correct folding of a specific motif within the ATP binding domain.

Considering that the ribosome exit tunnel can accommodate between 30 to 72 amino acids, the translation of the codon at position 3435 could coincide with the emergence of either the Q-loop or the Walker-A motifs from the ribosome exit tunnel (as depicted in Fig 8B). Notably, the Walker-A motif contains the ATP binding site, underscoring its functional importance.

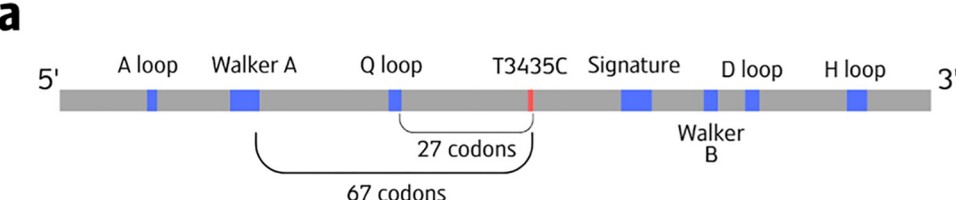

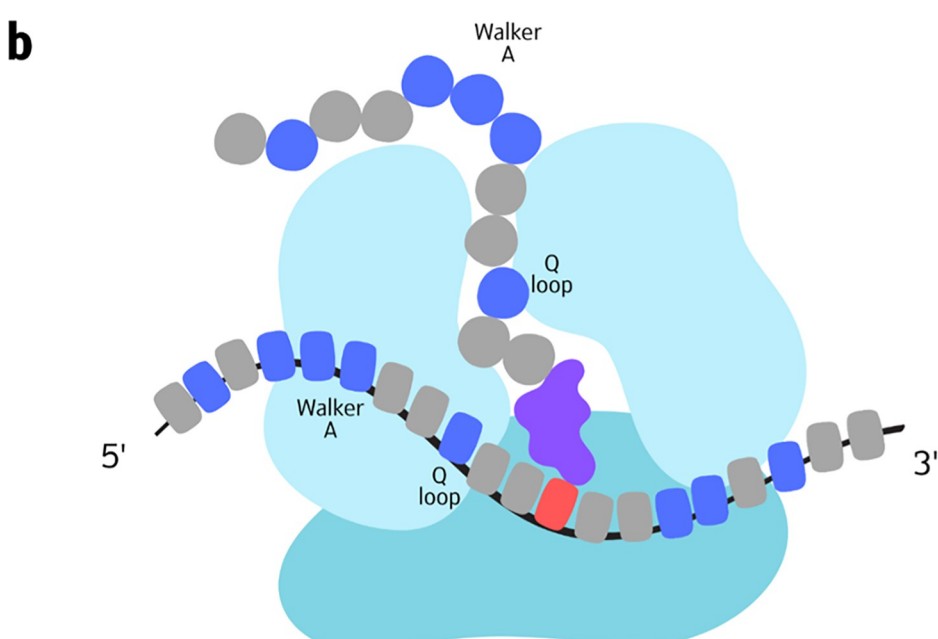

**Fig 8. An Illustration of the second ATP binding domain of *MDR1* and its translation.** a) An illustration of the structure of the second ATP binding domain of *MDR1*. The domain has a structure of a P-loop NTPase fold, containing seven conserved motifs, shown in dark blue. The location of the *T3435C* variant is shown in red and the rest of the domain is depicted in grey. The distances between *T3435C* and the Q-loop and Walker-A motifs are also noted. b) An illustration of the translation of the second ATP binding domain. The emerging of the Walker-A motif from the exit tunnel is simultaneous with the decoding of the variant codon, possibly affecting the folding of the motif. The small and large ribosomal subunits are depicted in shades of light blue; The position of the variant (3435) in the mRNA sequence is highlighted in red, while the motifs are shown in dark blue and the remaining mRNA sequence in grey. The tRNA is illustrated in purple. Amino acids within the motifs are colored dark blue, and other amino acids are shown in grey.

The *T3435C* variant, which our analysis suggests may increase the local translation rate, could potentially reduce the time available for proper folding of these critical motifs. This alteration in the co-translational folding process might lead to conformational changes in these motifs, which could in turn affect the ATP binding affinity of p-gp.

## 3.6. Novel SNPs and mutations suspected to affect local mRNA folding

Notably, the applied methods are not specific to *MDR1* and can be used systematical-ly on major databases to find mutations that affect mRNA folding and may lead to various pathologies. In the following section we will report our results on ClinVar and TCGA.

ClinVar is a database that contains hundreds of thousands of variants that are associated with various pathologies and conditions based on clinical and experimental evidence. The variants in the database are classified as either benign, pathogenic, or somewhere along the spectrum. For each variant we created a z-score, indicating whether there is a conservation of extremely low/high MFE at the variant's position (see Methods). We then grouped the ClinVar variants according to their z-scores and examined the ratio of pathogenic and benign variants at each group (Fig 9A). We find that while the pathogenic/benign ratio for all ClinVar variants is 59/41, the ratio for the 0.1% of variants with highest z-scores is 84/16 ($p = 1.5*10^{-10}$). Generally, variants located at positions with more extreme z-scores show a significantly higher likelihood of being pathogenic.

TCGA is a database that contains, among other genetic and clinical information, millions of somatic mutations that are found in the cancerous tissue samples of about 11,000 patients. Some mutations are cancer drivers while others are passengers; some initiate and progress the cancer while others are caused by it. While we do not have pathogenic and benign labels for TCGA variants, we can use genomic population databases to further validate the model. The 1000 Genomes project mapped the genetic variation of healthy individuals. Only variants that have an allele frequency larger than 1%, named polymorphisms, are present in the database.

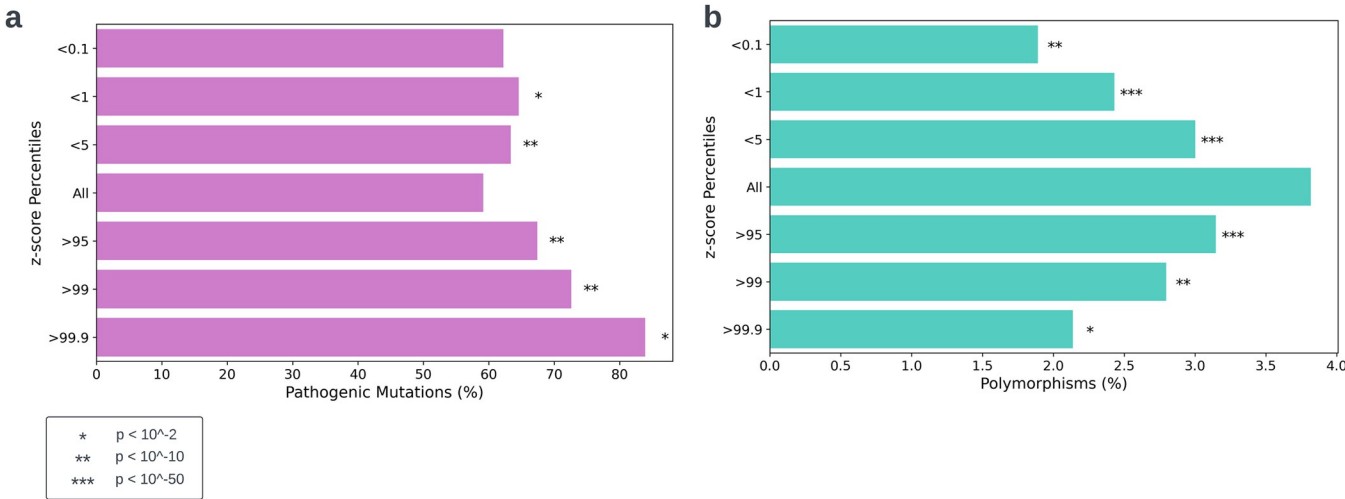

**Fig 9. Validation of MFE z-scores using ClinVar and TCGA.** a) ClinVar variants grouped by z-scores. The z-score indicates how extreme is the MFE at the variant position. The y- axis denotes the groups, "All" being all ClinVar variants and ">99.9" being the 0.1% of variants with the highest z-score. The x-axis denotes the percent of pathogenic variants in the group. Altogether, variants located at positions with more extreme z-scores show a higher likelihood of being pathogenic. b) TCGA variants grouped by z-scores. The z-score indicates how extreme is the MFE at the variant position. The y- axis denotes the groups, "All" being all ClinVar variants and ">99.9" being the 0.1% of variants with the highest z-score. The x-axis denotes the percent of variants that are polymorphisms that can be found in the 1000 genomes database. Variants located at positions with more extreme z-scores show a lower likelihood of being prevalent in the general population.

Our assumption is that TCGA variants that are also present in a general population database, such as 1000G, are less likely to be a cancer driver. While this principle surely does not apply to every single polymorphism, it serves as a reasonable overall assumption for distinguishing potential cancer drivers from benign polymorphisms. Similarly to the ClinVar analysis, we grouped the TCGA variants according to the z-scores of the positions where they reside. Then, we examine the percent of polymorphisms at each group (Fig 9B). We find that variants located at positions with more extreme MFE z-scores are significantly less likely to be present in the 1000G database, meaning that they are less prevalent in the population and are potentially more deleterious. Demonstrating that variants which reside in positions with conserved extreme MFE scores are more pathogenic and less frequent in the common population, suggests that this inquiry should be further pursued.

We searched both databases for specific variants whose characteristics suggest they can affect CTF. These are variants that are inside or adjacent to structural domains, are in positions with conserved extreme z-scores and cause a large change in MFE (see Methods for more details). We found 417 and 164 variants for ClinVar and TCGA respectively (S3 and S4 Tables). Analysis of TCGA variants revealed a missense to synonymous mutation ratio of 1.9:1. The most extreme MFE z-scores are -20 and +11.6 (Fig 10A), and the change in MFE caused by any variant ranges from -3.7 to 4.2 kcal/mol (Fig 10B). The most prevalent variant pan-cancer is a missense mutation that was identified in the cancerous tissues of 1,454 patients, representing 13% of all TCGA cancer patients across the 33 cancer types examined. The most prevalent variant in a specific cancer type is detected in over 30% of all Uveal Melanoma (UVM) patients (Fig 10D). When reviewing the genes in which the potentially CTF-altering variants reside compared to all TCGA mutations, we detected a 2.3-fold enrichment of known cancerous genes (Fig 10E). For meta-data of ClinVar potentially CTF-modifying variants, see S7 Fig.

Functional enrichment analysis (see Methods) indicates that genes with CTF-affecting variants in both TCGA and ClinVar are often related to enzymatic activities (for example "catalytic activity" and "hydrolase activity") and to cellular regulation processes (many terms related to nucleotide binding and GTPase activity). ClinVar variants additionally related to developmental and morphogenetic processes (see S5 and S6 Tables).

Delving into specific examples, a ClinVar variant suspected to affect CTF is *G1923A* in the *SMC1A* gene. *SMC1A* encodes for a protein that is part of the cohesin complex and takes part in chromosome segregation and in DNA repair [71]. *SMC1A* has ATP binding domains at its N-terminus and C-terminus and a single hinge domain (Fig 11D). The hinge domain allows the protein's flexibility and binding with other proteins, especially *SMC3*, to establish the cohesin complex (Fig 11A) [71]. *G1923A* is a rare synonymous variant that resides within the hinge domain (*rs782123095*). According to ClinVar, it is associated with Cornelia de Lange syndrome (CDLS). CDLS is a genetic disease that has a variable phenotype, including facial deformations and developmental delays [72]. About 5% of CDLS patients have mutations in the *SMC1A* gene, in various positions across all domains [73–75]. However, the impact and biological mechanism of *G1923A* remains unclear. We find that *G1923A* is in a region of conserved low MFE (Fig 11B) and causes a large and significant increase in MFE (Fig 11C). We hypothesis that this change modifies the CTF trajectory of the hinge domain, changing its conformation and impacting the interaction of *SMC1A* with other proteins.

Another example of a variant we hypothesize to effect CTF is *C732A* in the *RBFOX2* gene. *RBFOX2* is a splice factor that plays a key role in regulation of alternative splicing (illustrated in Fig 11E) [76]. It binds to various pre-mRNA, promoting or repressing exon inclusion, and can also interact with other splice factors and proteins. *RBFOX2* is known as a modulator of cells' epithelial to mesenchymal transaction (EMT) and metastasis [77–79]. *C732A*

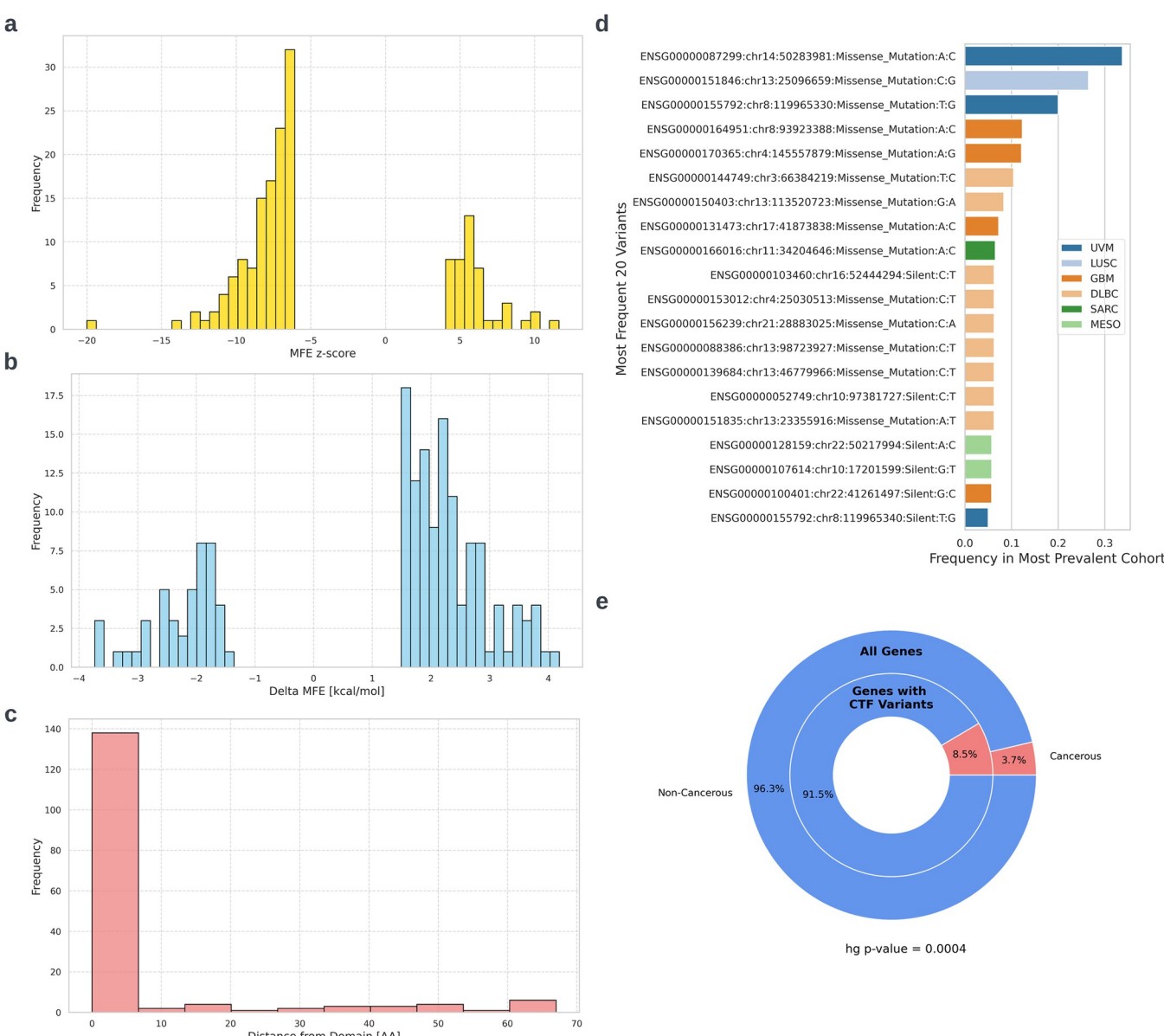

**Fig 10. Meta-data of potentially CTF-modifying variants in TCGA.** a) Distribution of MFE z-scores in the positions where the variants reside. b) Distribution of the change in MFE caused by the variants. c) Distribution of variants' distance from a structural domain. d) Frequency of top 20 variants in their most prevalent cancer type. UVM–Uveal Melanoma, LUSC–Lung Squamous Cell Carcinoma, GBM—Glioblastoma, DLBC–Lymphoid Neoplasm Diffuse Large B-cell Lymphoma, SARC—Sarcoma, MESO–Mesothelioma. e) Enrichment of cancerous genes among the genes with potential CTF modifying variants. The outer ring refers to all human protein coding genes with variants on TCGA while the inner ring refers to genes in which we found potential CTF modifying variants in TCGA. Red represents genes known to be related to cancer (TSGs and oncogenes) while blue represents genes not currently associated with cancer. The significance of the enrichment is calculated using a hyper-geometric p-value.

(*rs573361634*) is a synonymous variant in the ATP binding domain (ABD) of *RBFOX2* (Fig 11H). Though it is rare in the healthy population its frequency in the TCGA pan-cancer cohort is 0.2%. Moreover, 4% of Skin Cutaneous Melanoma patients carry this mutation. We find that *C732A* is in a region of conserved low MFE (Fig 11F) and causes a large and significant increase in MFE (Fig 11G). We hypothesis that this change raises the local translation speed at this slowly conserved region, affecting the CTF and thus the conformation of the binding

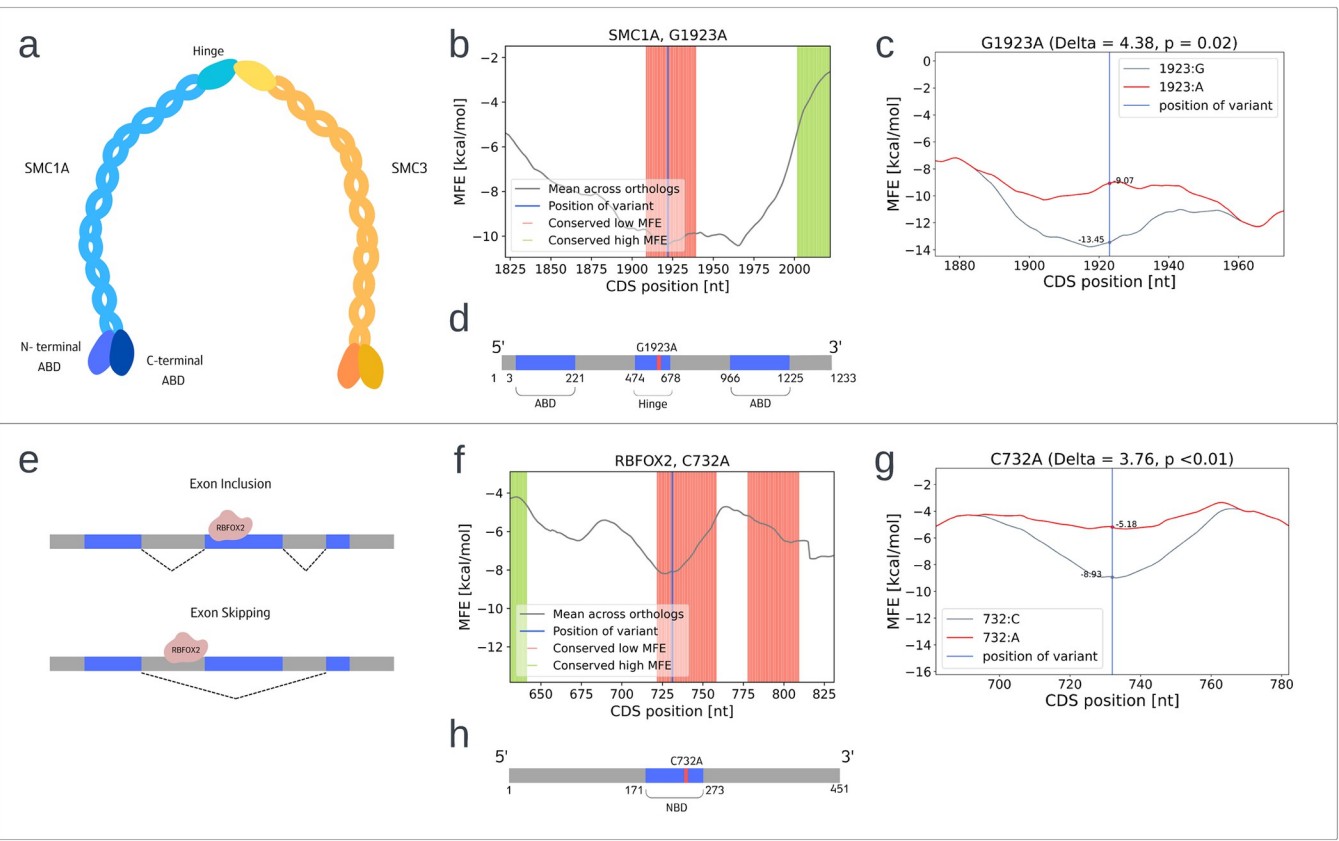

**Fig 11. Examples for potentially CTF modifying pathogenic variants.** a) Illustration of *SMC1A* function. Interact with *SMC3* and other proteins to create the cohesin complex and trap sister chromatids within it. *SMC1A* and *SMC3* interact through the hinge domain of both proteins. b) *G1923A* resided in a region of conserved low MFE. Description of the graph is the same as in Fig 7. c) *G1923A* caused a significant increase in local MFE in its vicinity. Description of the graph is the same as in Fig 2. d) Structural domains of *SMC1A*. The ATP binding domains and the hinge domain are depicted in blue, the variant in red and the rest of the residues in grey. e) Illustration of *RBFOX2* function. *RBFOX2* modulates splicing according to its binding location relative to the exons of the pre-mRNA. In the top part of the figure all three exons are included in the processed mRNA. In the bottom part, the middle exon is skipped. f) *C732A* resided in a region of conserved low MFE. g) *C732A* caused a significant increase in local MFE in its vicinity. h) Structural domains of *RBFOX2*. The nucleotide binding domain is depicted in blue, the variant in red and the rest of the residues in grey.

domain of *RBFOX2*. This change potentially impacts the binding of the splice factor to its substrates and the EMT pathway.

## 4 Discussion

Our study proposes a framework for analyzing the effects of genetic variants using computational approaches. This framework is especially advantageous for aspects of gene expression that are challenging to measure experimentally, such as CTF and translation dynamics. By leveraging predictive models and various measures, we systematically assess the impact of *MDR1* variants on the main phases of gene expression, providing a more detailed understanding of their potential roles in cancer prognosis and drug resistance.

Traditional studies have yielded inconsistent results regarding the impact of *MDR1* variants *T1236C*, *T2677G*, and *T3435C* on mRNA and protein expression, p-gp function, and patient outcomes. These inconsistencies are likely due to variations in study design, including differences in cohort size, patient ethnicity, tumor type, and chemotherapy regimens. By analyzing the gene expression process, we highlight what we believe to be the more reasonable effects of these variants and support some major previous studies.

A key contribution of this study is the development of a novel method to identify variants that are predicted to affect mRNA folding. This approach is particularly relevant for *MDR1*, as we find all three variants to significantly affect mRNA folding. This is predicted to cause an increase in the local translation rates in the vicinity of the variants, possibly leading to changes in the CTF trajectory in the case of *T3435C*.

Research conducted by Kimchi-Sarfaty et al. [21] revealed that the *T3435C* variant, when occurring in conjunction with either *T1236C* or *T2677G*, alters the substrate specificity of the p-gp protein. They hypothesized that *T3435C* modifies CTF (and therefore protein conformation and function) because it increases the local translation rate in a slowly translated region, important for correct CTF. They supported their hypothesis by identifying a group of infrequently occurring codons near *T3435C*, observing that this variant changes a rare codon to a common one. Building upon these findings, Fung et al. further investigated the effects of these variants on *MDR1* function [20]. Their study employed a more physiologically relevant model, utilizing cell lines that stably express human p-gp, in contrast to the transient high-expression system used previously. Consistent with Kimchi-Sarfaty's results, they observed that p-gp strains carrying different haplotypes (CGC, TTT, and TTA) exhibited distinct conformations and altered efflux patterns for several p-gp substrates in the presence of inhibitors. Notably, while no changes in ATPase activity were detected, the variants significantly influenced p-gp stability. The TTA and TTT haplotypes demonstrated a significantly extended half-life compared to CGC. These results underscore that the conformational changes induced by these variants not only affect substrate specificity but also impact protein stability, further elucidating the complex consequences of synonymous mutations in MDR1. Another study by Gottesman and Fung explores the potential impact of *T3435C* on the folding of the Q-loop and Walker-A motifs through CTF changes [80]. They discuss various factors that can influence the rate of protein synthesis, including tRNA abundance and mRNA secondary structure, but do not propose a specific hypothesis regarding which factor is affected by *T3435C*. While our analyses align with the conclusions of these studies, we provide new insights into the variant's mechanism of action; we demonstrate that *T3435C* significantly increases the MFE in a region of conserved low MFE. Conversely, the increase it causes to CAI and tAI is minor and not statistically significant. Thus, we hypothesize that the dominant factor is the effect of *T3435C* on mRNA secondary structure, likely increasing the local translation rate and altering the CTF trajectory of the ATP binding site. A review paper by Tsai et al. offers insight about synonymous variants and their effect on folding trajectories [81]; they emphasize that the p-gp variants do not cause a misfolded, non-functional protein but rather an alternative protein. This is also supported by the commonality of the haplotypes in the general population. They propose that changes in local translation rates, induced by these variants, result in slight alterations to the protein folding trajectory. This leads to close but distinct local minima in the protein folding landscape, potentially representing an evolutionary mechanism for expanding protein functionality.

When examining other gene expression aspects, such as mRNA expression or post-transcriptional modifications, we do not find significant associations. Moreover, though all three variants are predicted to increase the local translation rates, we do not predict them to influence protein abundance. These results strengthen our hypothesis that the mechanism of action of these variants is related to their effect on mRNA folding. Though *T1236C* and *T2677G* do not seem to modify CTF in a critical location, they could have many other implications due to their effect on mRNA folding. Variants that alter mRNA folding may not only impact CTF, but also may contribute to pathogenicity through other mechanisms; for example, they can influence mRNA stability [82], regulate splicing at intron-exon junctions [83], and modulate both translation initiation [84] and elongation kinetics [63,85].

Expanding our investigation, we show that there are hundreds of disease related variants in TCGA and ClinVar that are predicted to impact the folding in conserved genomic positions, similarly to *T3435C*. These variants were found enriched in genes associated with enzymatic activities and cellular regulation processes. Translation-dependent regulation of post-translational protein arginylation mediated by synonymous codon usage has been demonstrated for the purine nucleotide biosynthesis enzyme *PRPS2* [86]. It is possible that this phenomenon tends to appear in many additional enzymes and thus it is affected by mutations and SNPs that modulate mRNA folding.

One of the most compelling questions concerns how the three variants impact patient survival. Our findings, based on the clinical information of TCGA patients of all cancer types, suggests that *T1236C* is associated with decreased overall survivability. Though there have been many contradictive previous findings on the matter, our outcome supports some major previous studies. Chen et al. conducted a meta-analysis comprising 3,320 patients from 15 studies, determining that individuals with a TT genotype at position 1236 showed improved survival [38]. Additionally, Johnatty SE et al. examined 4,616 ovarian cancer patients from the Ovarian Cancer Association Consortium (OVAC) and TCGA, that have received chemotherapy treatments [87]. They found a marginal association of *T1236C* with worse overall survival of patients with nil residual disease and did not find association between *T2677G* or *T3435C* and survival parameters. Another question arising from this paper regards *T3435C*; if it impacts the conformation of MDR1, why doesn't it affect patient survivability? We hypothesize that the survival analysis may be too broad. Not all chemotherapy agents are p-gp substrates, and even within the substrates some might be more sensitive than others to the modification. If the cohort size had enabled conducting separate analyses according to the chemotherapy regimen, we believe a significant correlation would have been found for some chemotherapy agents. Additionally, cancer type likely serves as a significant confounding factor in this analysis. Ideally, each cancer type would be analyzed independently, but limited cohort sizes made this impractical. By grouping related cancer types (S5 and S6 Figs), we aimed to balance the need for specificity with the necessity of maintaining statistical power.

Our computational predictions regarding mRNA expression and protein abundance align with several key experimental studies in the literature, though the overall experimental evidence remains mixed. Studies by Kimchi-Sarfaty et al. and Fung et al. found no significant changes in mRNA and protein levels in their cell models, supporting our in silico findings. Similar results were reported by Gow et al. and Salama et al., who observed no significant effects on mRNA expression or protein abundance. However, it's important to note that some studies have reported conflicting results, finding variant-dependent changes in expression levels under specific conditions or in particular cell types. These contradictory findings underscore the complexity of gene regulation and the potential context-dependency of variant effects, highlighting the need for comprehensive experimental validation of computational predictions across different cellular contexts and conditions.

When reviewing the analyses described in this study it is important to acknowledge the limitations of our computational approach. For some phases of gene expression (such as transcription) multiple predictive models are available, each potentially yielding slightly or significantly different results, thereby impacting the conclusions drawn. Additionally, there are also gene expression aspects which were not accounted for such as post-translational modifications, which could play significant roles in mediating the effects of *MDR1* variants. Another potential caveat is that computational models are inherently limited by the quality and completeness of the input data. The use of TCGA data, for example, presents challenges related to data heterogeneity and potential biases. The small cohorts of carriers of the variants (26

patients or less) are a major challenge; when larger cohorts are available the analyses should be reconducted and patients should be split according to chemotherapy regimens.

It is important to note that our computational analyses of both *MDR1* and the TCGA and ClinVar datasets generate hypotheses rather than evidence. These predictions offer valuable insights and directions for future research but require experimental validation to confirm their biological relevance.

When feasible, future studies should include experimental assays to measure translation rates, protein levels, and functional assays of p-gp activity in cells expressing the MDR1 variants. This would provide direct evidence supporting the computational predictions and further elucidate the mechanisms through which these variants influence cancer progression and drug resistance.

In conclusion, our study underscores the potential of computational approaches to provide deep insights into the functional consequences of genetic variants. By integrating multiple predictive models and data sources, we can systematically explore the complex regulatory mechanisms of gene expression. While acknowledging the limitations and the need for experimental validation, our framework represents a significant step forward in understanding the molecular underpinnings of MDR1 variants and their impact on cancer biology.

## Supporting information

**S1 Fig. Main phases of gene expression.** Transcription: A region of the double helix of the DNA is unwound and a pre-mRNA sequence is transcribed using one of the DNA strands as templet. Splicing: The pre-mRNA is edited and introns are removed from the it. Additionally, a 5' cap and 3' poly-A tail are added to the mRNA. Translation: The ribosome synthesizes a protein according to the mRNA templet. During the translation process the nascent protein initiates the formation of secondary and tertiary structures. mRNA degradation: mRNAs undergo degradation and are broken to smaller fragments, mainly by ribonucleases. (PNG)

**S2 Fig. Comparison of MDR1 expression in TCGA patients with and without the haplotypes.** a) T1236C & T2677G; b) T1236C & T3435C; c) T2677G & T3435C;. d) T1236C, T2677G & T3435C. Red dot represents the mean MDR1 expression level of carriers, while violin plots depict the distribution of mean MDR1 expression levels among 100,000 randomly chosen non-carriers. Both groups are matched in size for accurate comparison. (PNG)

**S3 Fig. Kaplan-Meier survival curves of the carriers and non-carriers of haplotypes.** a) T1236C & T2677G; b) T1236C & T3435C; c) T2677G & T3435C;. d) T1236C, T2677G & T3435C. Left column: overall survival. Right column: progression-free survival. (PNG)

**S4 Fig. Overall survival curves of carriers vs. non-carriers of TCGA variants that are in high correlation with T1236C.** These variants are detected in 75% or more of T1236C positive patients. (PNG)

**S5 Fig. Overall survival curves of carriers vs. non-carriers of the three variants, stratified cancer types.** Top row–inflammatory cancers. Middle row- proliferative cancers. Bottom row-metabolic cancers. Categories are described in S1 Table. (PNG)

**S6 Fig. Progression-free survival curves of carriers vs. non-carriers of the three variants, stratified cancer types.** Top row–inflammatory cancers. Middle row- proliferative cancers. Bottom row- metabolic cancers. Categories are described in S1 Table.
(PNG)

**S7 Fig. Meta-data of potentially CTF-modifying variants in ClinVar.** a) Distribution of MFE z-scores in the positions where the variants reside. b) Distribution of variants' distance from a structural domain c)Distribution of the change in MFE caused by the variants. d) Distribution of disease categories associated with ClinVar variants potentially modifying CTF. Variants not associated with any condition or disease were excluded.
(PNG)

**S8 Fig. Illustration of the MSA randomization methods.** a) Original MSA of five orthologous sequences and amino acids. Each color represents a different amino acid. Synonymous codons are represented by numbering- For example, K1 and K2 are the two synonymous codons that encode for K. lines represent gaps in the MSA. b) Vertically permuted MSA. We shuffled the synonymous codons within each column in the original MSA that had a dominant amino acid. c) Horizontally permuted MSA. We randomly choose pairs of columns with the same dominant amino-acid and swap between the synonymous codons of these columns, keeping them in the same row.
(PNG)

**S9 Fig. Illustration of the per-position, averaged across orthologs MFE scores.** a) An illustration of the MFE scores mapped to the original (top) and randomized (bottom) MSAs. Red hues illustrated strong folding (more negative MFE scores) and green hues illustrates weak folding (less negative MFE scores). b) Taking the MFE scores of all randomizations for a single MSA position enables that calculation of the mean and standard deviation of the random MFE, which is then used to calculate a z-score for the position.
(PNG)

**S1 Table. Cancer type clusters**
(DOCX)

**S2 Table. change in MDR1 expression when stratifying patients according to cancer types.** Each cell displays the MDR1 fold change and the corresponding p-value in parenthesis.
(DOCX)

**S3 Table. ClinVar variants potentially affecting CTF.** The table contains characteristics of the mutation such as distance from closest structural domain, the z-score, the change it causes to the MFE and its association with a disease or phenotype.
(TSV)

**S4 Table. TCGA variants potentially affecting CTF.** The table contains characteristics of the mutation such as distance from closest structural domain, the z-score, the change it causes to the MFE and its frequency in different TCGA cohorts.
(TSV)

**S5 Table. Results of Functional enrichment analysis of potentially CTF affecting variants in ClinVar.**
(XLSX)

**S6 Table. Results of Functional enrichment analysis of potentially CTF affecting variants in TCGA.**
(XLSX)

**S1 Text. Methods for the co-translational folding model.**
(DOCX)

## Acknowledgments

We thank Yoram Zarai, Alma Davidson, Sanjit Batra and Yun S. Song for their contributions in different analyses of this study.

## Author Contributions

**Conceptualization:** Tamir Tuller.

**Data curation:** Tal Gutman, Tamir Tuller.

**Formal analysis:** Tal Gutman.

**Funding acquisition:** Tamir Tuller.

**Methodology:** Tal Gutman, Tamir Tuller.

**Supervision:** Tamir Tuller.

**Validation:** Tal Gutman, Tamir Tuller.

**Visualization:** Tal Gutman, Tamir Tuller.

**Writing – original draft:** Tal Gutman, Tamir Tuller.

**Writing – review & editing:** Tal Gutman, Tamir Tuller.

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
