## [Decision Letter · Decision Letter 0]

7 Oct 2024

Dear Dr. Tuller,

Thank you very much for submitting your manuscript "Evidence that MDR1 Variants affect cancer cells via their effect on mRNA folding" for consideration at PLOS Computational Biology. As with all papers reviewed by the journal, your manuscript was reviewed by members of the editorial board and by several independent reviewers. The reviewers appreciated the attention to an important topic. Based on the reviews, we are likely to accept this manuscript for publication, providing that you modify the manuscript according to the review recommendations.

Sincerely,

Mohammad Sadegh Taghizadeh, Ph.D.

Academic Editor

PLOS Computational Biology

Ilya Ioshikhes

Section Editor

PLOS Computational Biology

Reviewer's Responses to Questions

**Comments to the Authors:**

Reviewer #1: 

This paper uses computational approaches to re-explore the effect of three specific single nucleotide polymorphisms which occur commonly in the ABCB1 (MDR1 or ABCB1) gene. There have been many studies relating these single polymorphisms and combinations of them to clinical responses to chemotherapy (with conflicting results), and several papers using in vitro models (not all referenced here) showing that the synonymous polymorphism T3435C affects substrate specificity of the ABCB1 transporter through mechanisms related to protein and/or mRNA folding. Although this polymorphism is predicted to change the folding of mRNA, neither the previous papers nor this one provides direct evidence of such an effect. Thus, the title of the paper is misleading. Given that this is an area that has achieved a great deal of attention over many years, does this paper add additional information to the existing literature?

1. The data in Fig. 1 based on clinically available data indicate that the three polymorphisms under study are not associated with statistically significant changes in expression of ABCB1. This is one of the most definitive demonstrations in the literature of this observation.

2. Since polymorphisms in theory could affect mRNA splicing, and many examples have been found in other proteins of effects on splicing of SNPs, the lack of association with splicing variants is useful and novel information.

3. The analysis of codon usage is interesting and applicable to other systems.

4. The data showing a potential association of T1236C with patient outcomes is more problematic since it isn’t combined with detailed information about the tumors, their chemotherapy (?substrates transported by ABCB1), or toxicity information (a major function of ABCB1 is handling of drugs in the GI tract, kidney, and liver). In fairness to the authors, they indicate that this association needs further analysis including tumor and drug stratification.

5. The prediction that T3435C increases the local rate of translation suggests that translation rate could affect kinetics of protein folding at bottlenecks and, in theory, might explain effects on ABCB1 protein function. This is interesting, though in the absence of experimental verification, is hypothetical.

6. The authors apply their analysis to many other diseases resulting from genetic alterations and make several predictions about association of specific polymorphisms with severity of disease and/or presentation. Absent specific experimental verification, it is difficult to know whether these predictions are accurate and whether they represent a self-fulfilling hypothesis or can actually predict outcomes.

7. In the Discussion the authors refer to what appears to be a very relevant paper by Gottesman and Leung Fung as reference [76] but there are only 20 references in this paper and this paper does not appear in the reference list. Additional relevant references which should be discussed are attached.

Given the several new contributions this paper makes to the literature, especially the novel way of predicting “translatability” this contribution does appear to be significant. However, there should be more reference and Discussion of the existing literature which addresses the same issue (see above, point #7). In addition, the authors confuse computational prediction with experimental verification, including in the title “Evidence that MDR1 Variants affect cancer cells via their effect on mRNA folding.” There is no such “evidence” in the paper. The title should be “Computational Prediction that MDR1 Variants affect cancer cells via their effect on mRNA folding.”

Reviewer #2: 

The review has been added as an attachment

Reviewer #3: 

I will try to be brief The paper is excellent The authors studied the MDR1 T1236C, T2677G, and T3435C polimorfisms and the variants for the significantly changes the mRNA folding in their vicinity. This change in mRNA structure is predicted to increase local translation elongation rates, but not to change the protein expression Importantly the increased translation rate near T3435C is predicted to affect the protein’s co-translational folding trajectory in the region of the second ATP binding domain. This potentially impacts protein conformation and function.

The importance of this paper besides of being very solid is that the authors have used a computational approach to show molecular mechanisms of MDR1 variants and their potential impact on cancer prognosis and treatment resistance. But this is the tip of the iceberg on how this approach could be used for other proteins Last they have demonstrated the importance to identify mutations affecting mRNA folding in pathology.

I have the suggestion that the authors could improve this paper to explain why they have only showed the Kaplan Meier only for overall survival but not for the failure-free survival...

**Have the authors made all data and (if applicable) computational code underlying the findings in their manuscript fully available?**

Reviewer #1: **No: **Cannot evaluate. They refer to a technique not previously published and not described in enough detail to allow replication.

Reviewer #2: Yes

Reviewer #3: Yes

PLOS authors have the option to publish the peer review history of their article (what does this mean?). If published, this will include your full peer review and any attached files.

Reviewer #1: No

Reviewer #2: No

Reviewer #3: No

Figure Files:

Data Requirements:

Reproducibility:

References:

---

## [Decision Letter · Decision Letter 1]

29 Nov 2024

Dear Dr. Tuller,

We are pleased to inform you that your manuscript 'Computational Analysis of MDR1 Variants Predicts Effect on Cancer Cells via their Effect on mRNA Folding' has been provisionally accepted for publication in PLOS Computational Biology.

Best regards,

Mohammad Sadegh Taghizadeh, Ph.D.

Academic Editor

PLOS Computational Biology

Ilya Ioshikhes

Section Editor

PLOS Computational Biology

Feilim Mac Gabhann

Editor-in-Chief

PLOS Computational Biology

Jason Papin

Editor-in-Chief

PLOS Computational Biology

Reviewer's Responses to Questions

**Comments to the Authors:**

Reviewer #1: The authors have responded completely and appropriately to the issues raised by the reviewers.

Reviewer #2: I thank the authors for their extensive work in addressing the comments. The revised manuscript is much stronger and ready to be recommended for publication.

Reviewer #3: The authors have addressed all the reviewers comments

**Have the authors made all data and (if applicable) computational code underlying the findings in their manuscript fully available?**

Reviewer #1: Yes

Reviewer #2: Yes

Reviewer #3: Yes

PLOS authors have the option to publish the peer review history of their article (what does this mean?). If published, this will include your full peer review and any attached files.

Reviewer #1: No

Reviewer #2: **Yes: **Daniel del Valle Morales

Reviewer #3: No

---

## [Editor Report · Acceptance letter]

5 Dec 2024

PCOMPBIOL-D-24-01495R1 

Computational Analysis of MDR1 Variants Predicts Effect on Cancer Cells via their Effect on mRNA Folding

Dear Dr Tuller,

I am pleased to inform you that your manuscript has been formally accepted for publication in PLOS Computational Biology. Your manuscript is now with our production department and you will be notified of the publication date in due course.

With kind regards,

Zsofia Freund
